# Pressure induced enhancement of the magnetic ordering temperature in rhenium(IV) monomers

Christopher H. Woodall[1,2,*], Gavin A. Craig[3,*], Alessandro Prescimone[1], Martin Misek[4], Joan Cano[5,6], Juan Faus[5], Michael R. Probert[7], Simon Parsons[1], Stephen Moggach[1], José Martínez-Lillo[1,5], Mark Murrie[3], Konstantin V. Kamenev[2,4] & Euan K. Brechin[1]

Materials that demonstrate long-range magnetic order are synonymous with information storage and the electronics industry, with the phenomenon commonly associated with metals, metal alloys or metal oxides and sulfides. A lesser known family of magnetically ordered complexes are the monometallic compounds of highly anisotropic d-block transition metals; the 'transformation' from isolated zero-dimensional molecule to ordered, spin-canted, three-dimensional lattice being the result of through-space interactions arising from the combination of large magnetic anisotropy and spin-delocalization from metal to ligand which induces important intermolecular contacts. Here we report the effect of pressure on two such mononuclear rhenium(IV) compounds that exhibit long-range magnetic order under ambient conditions via a spin canting mechanism, with $T_c$ controlled by the strength of the intermolecular interactions. As these are determined by intermolecular distance, 'squeezing' the molecules closer together generates remarkable enhancements in ordering temperatures, with a linear dependence of $T_c$ with pressure.

[1] EaStCHEM School of Chemistry and Centre for Science at Extreme Conditions, The University of Edinburgh, David Brewster Road, Edinburgh EH9 3FJ, UK. [2] School of Engineering and Centre for Science at Extreme Conditions, The University of Edinburgh, Erskine Williamson Building, Peter Gurthrie Tait Road, Edinburgh EH9 3FD,UK. [3] WestCHEM School of Chemistry, University of Glasgow, University Avenue, Glasgow G12 8QQ, UK. [4] School of Physics and Centre for Science at Extreme Conditions, The University of Edinburgh, Erskine Williamson Building, Peter Gurthrie Tait Road, Edinburgh EH9 3FD, UK. [5] Departament de Química Inorgànica/Instituto de Ciencia Molecular (ICMol), Universitat de València, C/ Catedrático José Beltrán no 2, 46980 Paterna (València), Spain. [6] Fundació General de la Universitat de València (FGUV), Universitat de València, Valencia c/Amadeo de Saboya 4, 46010, Spain. [7] School of Chemistry, Newcastle University, Newcastle upon Tyne NE1 7RU, UK. * These authors contributed equally to this work. Correspondence and requests for materials should be addressed to J.M.-L. (email: F.Jose.Martinez@uv.es) or to K.V.K. (email: K.Kamenev@ed.ac.uk) or to E.K.B. (email: E.Brechin@ed.ac.uk).

Magnets are ubiquitous in modern society, employed in an enormous range of applications from information storage, biomedical imaging and cancer therapy to space research[1–5]. One of the main goals of modern academic and industrial research is device miniaturization, and a bottom–up or molecular approach to building components represents an attractive methodology[6,7]. The synthesis of molecules whose behaviour resembles that of classical bulk magnets has been an important challenge for several decades[8–17]. Here, the physical behaviour is, in part, governed by the magnetic anisotropy of the molecule, which in turn originates from symmetry and structure—factors that are a challenge to control via synthetic chemistry. An alternative way of harnessing and exploiting magnetic anisotropy, and other important factors such as the nature and strength of intra- and intermolecular exchange interactions, is through the use of pressure, since the latter can be used to modify intramolecular bond lengths, angles and metal geometries, and important intermolecular interactions such as H-bonds, C–H···$\pi$ and $\pi$···$\pi$ contacts, amongst others[18]. Magnetic anisotropy also plays a significant role in spin-canted systems[19], which behave as weak ferromagnets[10]. In these systems, magnetic order originates from the non-colinearity of neighbouring spin centres which are 'canted' at a particular angle ($\alpha$) with respect to each other[10]. Importantly the non-negligible intermolecular magnetic interactions can be modified by changing intermolecular distances, for example, making these distances shorter would be expected to increase the strength of the exchange and increase the ordering temperature, $T_c$ (ref. 20). An obvious way of achieving this is to exert hydrostatic pressure, and by combining high-pressure single-crystal X-ray crystallography and high-pressure SQUID magnetometry the exact relationship between changing structure and changing magnetic behaviour can be extracted[21].

Among suitable candidates for this type of study are mono-nuclear complexes based on the hexahalides of the Re$^{IV}$ (5d$^3$) ion[20,22,23], which are characterised by large magnetic anisotropies and significant intermolecular magnetic exchange[20]. Indeed, such species can display relatively strong dipolar exchange through Re–X···X–Re type contacts (X = halogen) which can result in spin canting[23–27]. DFT calculations reveal these exchange pathways arise as a result of the spin density from the metal ion being delocalized onto the peripheral atoms of the ligands[28–30], an effect not observed in analogous 3d metal complexes which do not order. Herein we report a high pressure study of the complexes [ReCl$_4$(MeCN)$_2$]·MeCN (1) and [ReBr$_4$(bpym)] (2) (bpym = 2,2'-bipyrimidine). 1 is a well-known starting material for the preparation of rhenium-based compounds. Curiously, although a straightforward synthetic method to 1 has been known since 1968 (ref. 30), and its crystal structure was recently solved[31,32], its magnetic behaviour has never been reported. The synthesis and crystal structure of 2 (ref. 33), together with electrochemical[33] and preliminary magnetic studies[25], are known. 1 and 2 exhibit magnetic ordering via spin canting. Here we demonstrate that a rather remarkable enhancement of $T_c$ can be achieved for both 1 and 2 through the application of modest pressures. A combined high-pressure structural and magnetic study reveals that shortening inter-molecular interactions result in a linear dependence of $T_c$ with pressure, and a complementary theoretical study extracts the changes in both the exchange interaction parameters and anisotropy of the Re$^{IV}$ ion that lead to this enhancement.

## Results

**High-pressure X-ray diffraction studies of 1 and 2.** The crystal structures of compounds 1 and 2 at ambient pressure have been

reported and discussed elsewhere[31–33]. However, it is useful to discuss certain features to understand the structural changes observed with hydrostatic pressure, and the possible effects that such changes may have on the magnetic properties. Both compounds crystallise in orthorhombic space groups (Pnma and P2$_1$2$_1$2$_1$ for 1 and 2, respectively) and, although not isostructural, they have several features in common that may aid the explanation of the unique magnetic behaviour observed at pressure. Both compounds exhibit a Re$^{IV}$ ion in a somewhat distorted octahedral environment, being bonded by two N atoms coordinated via the organic ligand(s) and four halide anions (Supplementary Fig. 1). Both display staggered zig-zag chains that propagate along the a axis of the unit cell via interhalide dispersive interactions (Fig. 1), with 1 forming a layered structure of chains of [ReCl$_4$(MeCN)$_2$] alternating with disordered MeCN molecules that occupy voids that run parallel to the chains. The chains observed in 2 pack to form sheets in a manner reminiscent of herring bone type structures (Supplementary Fig. 2).

On compression, the structures of 1 and 2 undergo significant contraction, reflected in a reduction in cell parameters and volume for both (Supplementary Tables 1 and 2, and Supplementary Data 1 and 2). The most noticeable effect is the reduction of the unit cell volume from 1381.11(8) Å$^3$ under ambient conditions to 1102.4(13) Å$^3$ at 4.30 GPa—a change of 20% for 1, and from 1402.5 Å$^3$ at ambient to 1202.3 Å$^3$ at 3.64 GPa—a change of 14% for 2. In both systems the principal component of

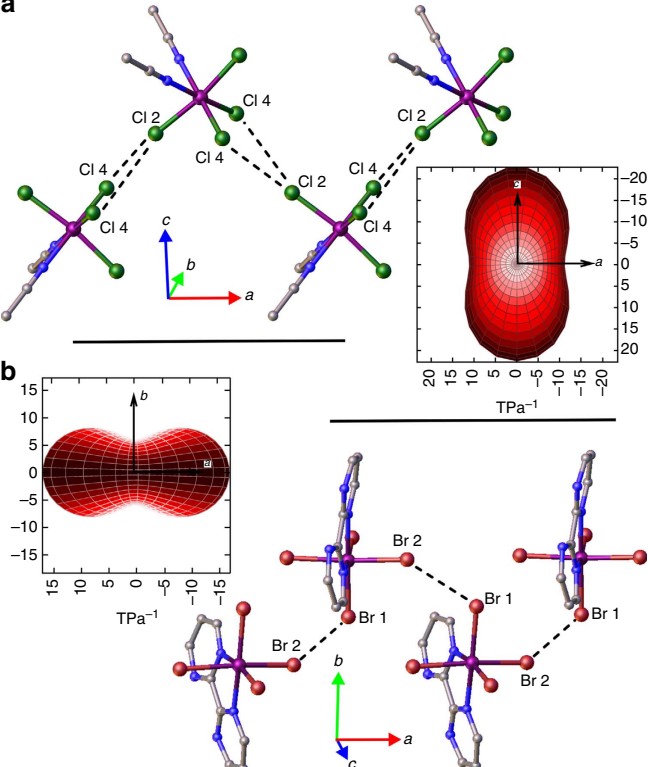

**Figure 1 | Structural comparison of 1 and 2 at high pressure.** Comparison of the packing between chains showing the shortest intermolecular Re–X···X–Re interactions in 1 (a) and 2 (b). Inserts: compressibility indicatrix of 1 and 2 generated from the high-pressure data showing the spatial orientation of strong-positive compressibility axes relative to their crystallographic axes and chain direction. In 1 the principal axis of compression lies perpendicular to the Re–X···X–Re chain, while in 2 it appears parallel to the equivalent interactions. The discepancy in behaviour is attributed to the continuous void occupied with MeCN that runs along the c axis in 1.

compression occurs down the *c* axis of the unit cell, and the contraction occurs in a way that compresses the structure into the voids occupied by the disordered solvent in **1** (Fig. 1 and Supplementary Table 3). An important point to note is the structural transformation of compound **2** between 1.93 GPa and 3.06 GPa from the ambient $P2_12_12_1$ orthorhombic phase to a previously unreported monoclinic phase with space group $P2_1$ (Supplementary Data 2). The transformation results in two molecules of **2** in the asymmetric unit opposed to the one molecule observed in the ambient phase.

The behaviour of the intermolecular Re–X···X–Re distances is of great importance in these systems[20]. Under ambient conditions there are two types of halogen–halogen interactions in **1** of very similar distances (Fig. 1 and Supplementary Table 4); Re–Cl(4)···Cl(4)–Re is 4.014(2) Å and Re–Cl(4)···Cl(2)–Re is 3.9217(13) Å. While the application of pressure does not seem to affect the former very much (3.912(14) Å at 4.30 GPa), the latter shortens by ∼ 0.6 Å to 3.362(11) Å. Compound **2** displays a similar reduction in interbromide distances, albeit smaller on average, with Re-Br(2)···Br(4)-Re changing by 0.31 Å from ambient to 3.64 GPa and reducing in a manner that is reflective of the more isotropic compression of the material (Supplementary Table 5).

Analysis of the significant intramolecular bond lengths for **1** and **2** reveals limited change with pressure for both compounds. The values of the Re–Cl and Re–N bond lengths in **1** are in agreement with those previously reported and they remain essentially constant with the different applied pressures (Supplementary Table 6). Similar behaviour is observed in **2** (Supplementary Table 7). However, the axial Cl(2)–Re(1)–Cl(3) angle distorts significantly from 'linearity' with increasing pressure (Supplementary Fig. 3) by ∼ 3.0° (**1**) and 4.3° (**2**) from that observed under ambient conditions, resulting in a significant distortion to the previously discussed octahedron. This distortion may have important consequences for the anisotropy of the Re[IV] ion (*vide infra*).

Both samples were subjected to complete compression–decompression cycles up to 4 GPa and were found to return to their original ambient pressure unit cell and space group with no significant changes in structure. To exclude the possibility that a low temperature structural transition drives modulation of the magnetic properties in **1** and **2** (*vide infra*), their crystal structures were solved at $T = 4$ K. On cooling, both materials were found to retain their room temperature structures, with the only differences arising from the expected thermal contraction (Supplementary Table 8 and Supplementary Fig. 4).

**High-pressure magnetic studies of 1 and 2.** To determine the impact that the pressure-induced structural modifications have on the magnetic properties of compounds **1** and **2**, HP SQUID magnetometry was employed. An ambient pressure magnetic study of **1** as measured on a sample in a gelatine capsule is given in Supplementary Note 1 and Supplementary Figs 5 and 6. High-pressure dc magnetic susceptibility measurements were performed on microcrystalline samples of **1** and **2** in a turnbuckle diamond anvil cell[34]. The ordering temperature for both compounds, $T_C$, was determined by field cooled and zero-field cooled measurements carried out under a field of 100 Oe, the temperature at which the remanent magnetization becomes non-zero. Figure 2 shows the field cooled–zero-field cooled curves as a function of temperature for compounds **1** and **2** at the various pressure points measured. At ambient pressure in the turnbuckle cell, compounds **1** and **2** display $T_C$ values of 6.5 and 18.0 K, respectively. For both complexes, the application of hydrostatic pressure leads to an increase in the size

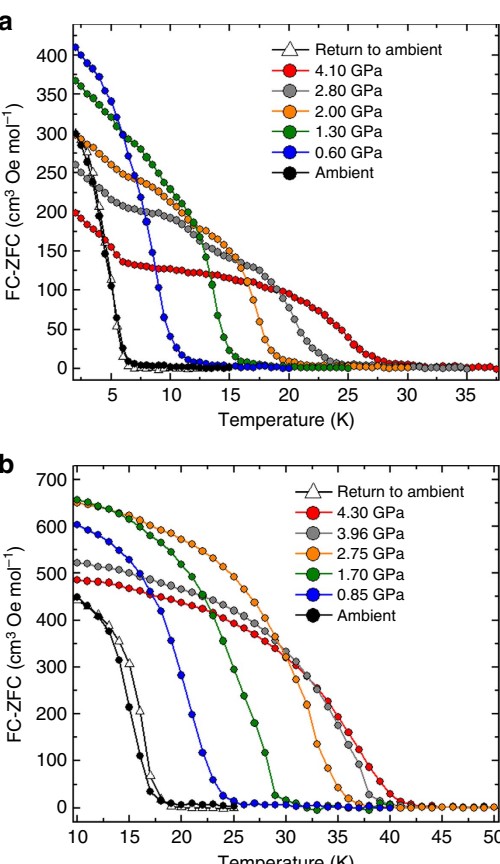

**Figure 2 | Pressure dependence of the magnetic properties of 1 and 2.** Plot of field cooled − zero-field cooled curves versus temperature for **1** (**a**) and **2** (**b**). In both compounds the $T_c$ value is shifted to higher temperatures with increasing pressure. Both compounds were measured under an applied field of 100 Oe. High-pressure data were collected in a turnbuckle DAC designed specifically for use in an MPMS magnetometer, while the return to ambient pressure were measured on the same samples in a gelatine caspule.

of the remanent magnetization measured, a phenomenon that was ascribed to the emergence of anisotropy in the case of the compound $Mn[N(CN)_2]_2$ (ref. 35). More remarkable, however, is the rise observed in the ordering temperature of **1** and **2** as the pressure is increased. At the highest pressures measured, $T_C$ reaches a value of 28 K at 4.10 GPa for compound **1**, and 42 K at 4.30 GPa for compound **2**. In the case of compound **1**, this represents a four-fold enhancement of the ordering temperature. The rate at which $T_C$ increases with pressure was determined to be ∼ 5.1 and 5.4 K GPa$^{-1}$ for **1** and **2**, respectively (Fig. 3). The phase transition observed for compound **2** above 1.93 GPa does not appear to affect the monotonic increase of the ordering temperature. These increases in $T_C$ are consistent with the contraction in the intermolecular halogen···halogen distances revealed by the high-pressure crystallographic study. To demonstrate that the effects of pressure on the magnetic properties of **1** and **2** are reversible, the pressure was released from the cell, the samples removed and their remanent magnetization measured in gelatine caspules. In both cases, a recovery of the ambient pressure $T_C$ is observed (Fig. 2). To analyse the exchange interactions in a quantitative manner, and to evaluate a possible change in the magnetic anisotropy of the distorted octahedral Re[IV] ion, we now turn to theory.

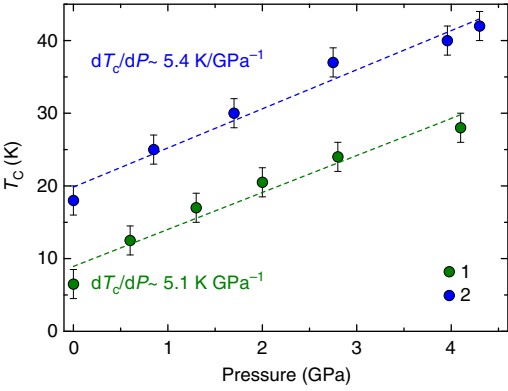

**Figure 3 | Enhancement rates of the magnetic ordering temperature for compounds 1 and 2.** The magnetic ordering temperature ($T_C$) versus applied pressure plot for **1** (green) and **2** (blue) shows a linear increase of $T_C$ values with applied pressure. For **1** this corresponds to 5.1 K GPa$^{-1}$, and for **2** this corresponds to 5.4 K GPa$^{-1}$. The error bars for each data point represent twice the size of the temperature measurement interval.

**Theoretical studies.** The first question to be answered by theory is if the necessary conditions to observe spin canting have been met, and thus we have employed density functional (DF) calculations to evaluate the magnetic exchange interactions between Re$^{IV}$ ions, and the evolution of this exchange with pressure. The only magnetic interactions that can exist in **1** and **2** are intermolecular in nature. Dipolar magnetic exchange is normally rather weak, and indeed for 3d metals even negligible. However, this is not the case for mononuclear Re$^{IV}$ complexes in which there exists strong $\pi$ spin delocalization from the Re$^{IV}$ ion onto the coordinated ligands, particularly to the donor atoms that are directly connected to the metal ion. The result is non-negligible magnetic interactions between spin densities on neighbouring molecules[20]. Figure 4 shows spin density maps for **1** and **2** which reveal large spin delocalization onto the chloride **1** and bromide **2** ions, and little to the peripheral regions of the organic ligands. Significant spin polarization is also seen on the nitrogen atoms of the acetonitrile and bipyrimidine molecules that are directly coordinated to the metal ion in **1** and **2**, respectively. The important intermolecular interactions would be expected to occur only between atoms exhibiting large spin densities.

Several intermolecular contacts were selected after a detailed inspection of the crystal packing of both compounds (Supplementary Figs 7 and 8) and DF calculations were performed on these moieties to estimate the magnitude and nature of the magnetic exchange ($J_i$) at varying pressures. The computed $J_i$ values are given in Supplementary Tables 9 and 10. As expected, these results show that the shortest pathways involving the chloride and bromide anions generate the most efficient magnetic exchange at all pressures ($J_{1-3}$ in **1** and $J_{1-2}$ in **2**). The magnitude of these interactions is indeed large enough to induce magnetic order. However for spin canting the easy-axis of magnetization of the exchange coupled metal centres must also be strictly non-colinear. Several factors can contribute to controlling the orientation of these axes, but there is little doubt that the presence of magnetic anisotropy is crucial. The axial ($D$) and rhombic ($E$) zero-field splitting (zfs) parameters for **1** and **2** were evaluated from CAS calculations, confirming very large axial magnetic anisotropy arising from second-order spin–orbit coupling (Supplementary Table 11). From these results it is also possible to determine $D$ tensor projections and the angles ($\alpha$) between projections on different Re$^{IV}$ ions (Supplementary Fig. 9). We can then compare

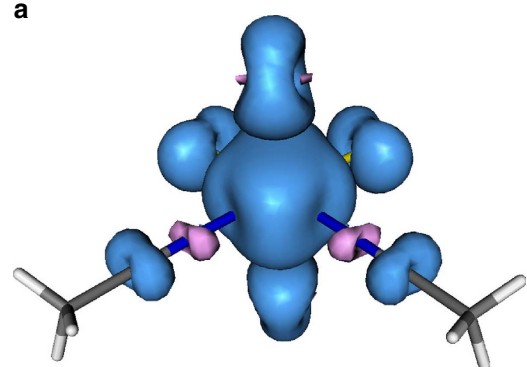

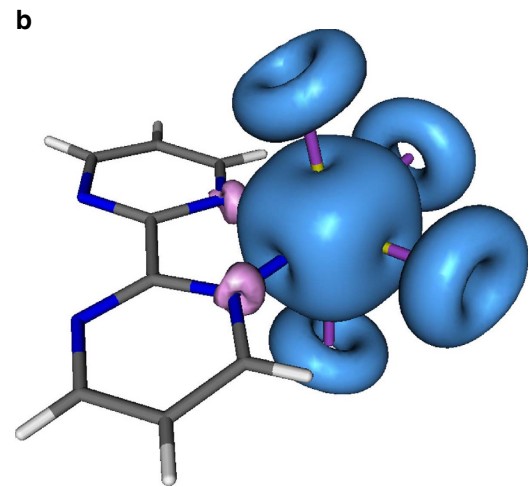

**Figure 4 | Spin densities calculated through DF for 1 and 2.** View of the calculated spin density for the $S = 3/2$ ground spin configuration of the [ReCl$_4$(MeCN)$_2$] (**a**) and the [ReBr$_4$(bipyrimidine)] (**b**) complexes in **1** and **2**, respectively. The isodensity surface corresponds to a cutoff value of 0.003 e bohr$^{-3}$. Blue and magenta isosurfaces correspond to positive and negative regions of spin density, respectively. Spin densities are largely centred on the metal ion, but with strong delocalization to the chloride and bromide ions. Non-negligible spin polarization (magenta) is observed on the nitrogen atoms of the N-donor ligands.

the theoretically derived $\alpha$ angles ($\alpha_{THEO} = 0$ and 7.3°) with those found experimentally ($\alpha = 2.0$ and 2.5° for **1** and **2**, respectively; see Supplementary Note 1) from the saturation value of the magnetization, or from the saturated magnetic susceptibility at the lowest temperature[25]. In **2**, the strongest magnetic exchange ($J_1 = -2.25$ cm$^{-1}$) leads to a theoretical $\alpha_{THEO} = 7.3°$, formed by the $z$ components of the $D$ tensor of two neighbouring [ReBr$_4$(bpym)] complexes, close to that extrapolated from experiment, and confirming non-colinearity. The situation in **1** is less straightforward since at ambient pressure three similar intermolecular magnetic interactions coexist, and inversion symmetry between neighbours should strictly define $\alpha = 0$. Experimentally, the presence of a non-zero $\alpha$ angle can originate from several sources, including non-isotropic exchange interactions and/or a difference in the structure/symmetry of the complex at high temperature where the X-ray structure was solved and low temperature where the magnetic order occurs. However, low temperature ($T = 4$ K) single-crystal X-ray diffraction at ambient pressure shows no significant change to the structure in **1** or **2**, ruling out the latter. To visualize if a non-zero value of the $\alpha$ angle can exist, a molecular dynamics simulation of **1** was performed with DF methods in the solid state at $T = 5$ K

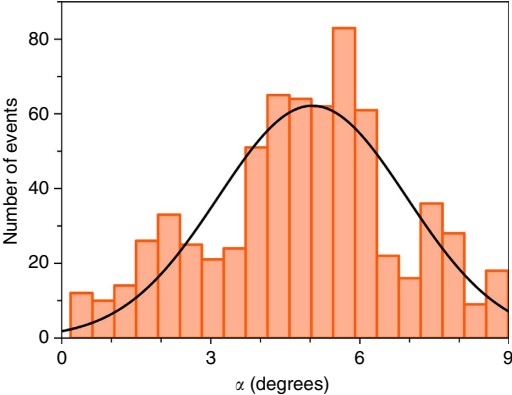

**Figure 5 | Histogram of the $\alpha$ value in 1.** Distribution of the $\alpha$ value obtained from molecular dynamics at $T = 5$ K. Bars show the number of events for each interval value and the solid black line is the best-fit to a Gaussian curve. A total of 1,000 events were recorded with a time length of 1 fs per step. A maximum in the distribution is observed around $\alpha = 5.0°$.

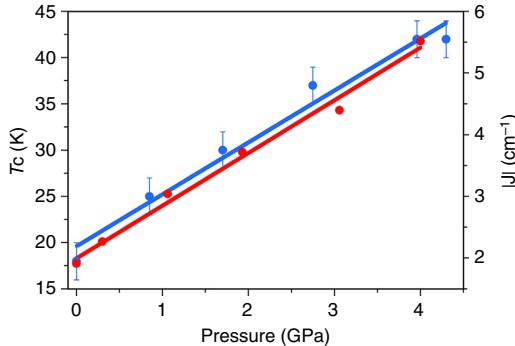

**Figure 6 | Magnetostructural correlations in 2.** Pressure dependence of the ordering temperature, $T_c$ (blue circles), and the strongest magnetic exchange, $J$ (red circles). The lines represent the linear best-fit. The magnetic exchange interactions were calculated on the experimental geometries for each experimental pressure determined. The data points above 3.5 GPa are above the pressure where the structural phase transition is seen crystallographically. The error bars for each data point of the experimentally derived $T_c$ are given.

from a pre-optimized structure. From NEVPT2 and CAS methods, an evaluation of the zfs parameters, including the orientation of the $D$ tensor, was made on all the geometries found in the molecular dynamics of the two neighbouring $[ReCl_4(MeCN)_2]$ units (**a** and **b**) involved in the $J_1$ magnetic exchange. The results are shown in Fig. 5 and Supplementary Figs 10 and 11. Very similar results were found from both NEVPT2 and CAS calculations, revealing non-negligible changes in the time evolution of the $\alpha$ angle and $D$ parameter for **a** and **b**.

Histograms of the time evolutions show that the values of both parameters follow a Gaussian distribution centred at $\alpha_{SIM} = 5.0°$ and $D_{SIM} = -26.0$ cm$^{-1}$. The $D_{SIM}$ value is very close to that obtained theoretically from the experimental crystal structure at ambient pressure ($D_{THEO} = -30.0$ cm$^{-1}$) and also to that obtained experimentally from the fit of $\chi_M T$ versus $T$ plot in the temperature range 20–300 K ($D_{EXP} = -37.1$ cm$^{-1}$; Supplementary Fig. 5). The non-zero $\alpha_{SIM}$ value supports the observed spin canting phenomenon in **1**. Supplementary Movies 1 and 2 allow us to visualize the changes in the geometries of **a** and **b** with time. Although these structural changes are not large, they are big enough to induce modifications in the $D$ parameter, and significant enough to allow the spin-canting phenomenon to occur. In short, the significant zfs allows the magnetization axes on neighbouring metal centres to be non-collinear, resulting in spin canting.

Anisotropy calculations were performed on the molecular geometries of **1** and **2** at higher pressures (Supplementary Note 2), and a summary of the results is given in Supplementary Table 11 and Supplementary Figs 12 and 13. Compound **2** is magnetically more anisotropic when external pressure is applied, whereas **1** becomes less anisotropic. However, the changes are rather small in both cases. The changes in the $D$ parameter are easily explained by changes to the molecular geometry, and in particular to distortions in the axial X–Re–X moiety. Only the spin–orbit part is significant here, with contributions from both the quartet and doublet excited spin states, the latter being the most important (Supplementary Note 2). Thus, changes to the octahedral geometries of the Re ions in **1** and **2** with pressure have changed their electronic structure energy diagrams, that is, the relative energies of the excited states have been modified and their contributions to the zfs have changed accordingly. However, the observed distortions are similar (and relatively small) in both complexes [3.0 (**1**) and 4.3° (**2**)] and the modification of $D$ in each case is not enough to explain the large changes in $T_c$.

On the other hand, significant changes in the magnitude of the intermolecular magnetic exchange interactions can considerably influence the magnetic behaviour. Thus, an increase in $T_c$ would be expected when the magnetic exchange constants become stronger. DF calculations reveal an increase in the magnitude of the $J_i$ parameters in **1** and **2** when external pressure is applied, in agreement with increasing $T_c$ in both systems. In **2**, the correlation between parameters is beautifully straightforward, external applied pressure produces a linear increase in $J_1$ and a linear increase in $T_c$ (Fig. 6). The analysis of **1** is again complicated by the presence of three similar $J_i$ values, although the overall trend seen above is repeated (Supplementary Fig. 14), and only the $J_i$ value at 4.3 GPa deviates from the expected correlation. We focus on the $J_1$ interaction. In **2** the $J_1$ magnetic exchange only involves the spin densities on neighbouring Br ions, the distance between which shortens with pressure. In **1**, $J_1$ again involves the halide $\cdots$ halide contact but also that between halide and both the N and C atoms of an acetonitrile molecule. In other words, three contacts are directly involved in the interaction. To complicate matters further, the spin densities on the C and N atoms exhibit different signs, and moreover, the intermolecular distances in these three contacts do not vary in the same manner with pressure (Supplementary Fig. 15). Thus, deconvoluting the individual contributions of these three simultaneous changes is non-trivial. From Supplementary Fig. 14 it is clear that the $T_c$ and $J$ values calculated for the highest pressure measured (4.3 GPa) deviate from the linear correlation. This can be attributed to the anomalously large carbon–nitrogen bond lengths in the coordinated acetonitrile molecule that are clearly very different to the values found at all other pressures (Supplementary Tables 12 and 13). This structural discrepancy is associated to the difficulties in obtaining structural data at such high pressures. The result is an underestimation of the strength of the magnetic exchange at this pressure.

## Discussion

While chemists have become accustomed to the use of high temperatures and high magnetic fields, the use of high-pressure techniques remains hugely under-exploited—particularly in molecular chemistry. This is surprising given that the physical properties of molecules are inherently linked to their structures and any modification of the latter will have important, and potentially game-changing consequences on the former. Perhaps

one reason is the oft-heard but incorrect assumption that the only consequence of applying pressure to molecules in crystals is the removal of interstitial void space. This is the obvious initial consequence, but when these voids are removed the molecules push against each other and it is at this point and beyond where novel physical properties emerge. Here, we have demonstrated the pressure-induced enhancement of the magnetic ordering temperatures of two spin-canted $Re^{IV}$ systems, and have correlated changes in the magnetic behaviour with changes in structure through the combined use of high-pressure single-crystal X-ray crystallography and high-pressure SQUID magnetometry. This high-pressure methodology has allowed unprecedented insight into the phenomenon in a step-by-step manner, observing remarkable reductions in unit cell volume, with the concomitant shortening of intermolecular distances resulting in a linear increase in $T_c$. DF-type calculations have allowed us to evaluate the magnetic exchange interactions ($J_i$) between the $Re^{IV}$ ions, and how these $J_i$ values are affected by the application of pressure. The $J_i$ values increase with increasing pressure on account of closer intermolecular contacts enforcing stronger exchange coupling between adjacent $Re^{IV}$ centres at each increased pressure. Pressure-induced enhancement of $T_c$ has been previously reported through SQUID magnetometry in 3D ferro- and ferrimagnets, and in weak ferromagnets containing transition metal ions[35–43], although in each case no structural information was forthcoming, preventing unequivocal explanation of the observed magnetic changes. The present study represents the first combined high-pressure single-crystal X-ray crystallography—high-pressure magnetism study of an ordered 5d molecule-based material. The changes in $T_c$ are quite remarkable and originate, in the main, from a simple shortening of intermolecular distances, which results in larger dipolar magnetic exchange. This opens the gateway to studying the pressure-dependent magnetic behaviour of the relatively large number of anisotropic, monomeric transition metal complexes which also order at low temperatures. One can imagine similarly impressive results in such systems, particularly for 5d metal ions on account of their increased magnetic anisotropy arising from their substantial spin–orbit coupling constants and the larger diffuseness of their magnetic orbitals when compared with those of the 3d and 4d ions[44]. Indeed the linear increase in $T_c$ with pressure seen here suggests that enhancement rates in other species may be much larger and that pressure could be employed to invoke transition from paramagnetism to LRMO.

Finally, we hope to have demonstrated the power, usefulness and potential of employing high-pressure techniques in molecular chemistry. Here we employed high-pressure X-ray crystallography and high-pressure SQUID magnetometry to reveal the relationship between structure and magnetic behaviour, but other high-pressure techniques also exist and remain largely unexplored. For example, high-pressure INS, Raman, infrared/ultraviolet–visible/near-infrared and EPR spectroscopies are all available and offer the chemist a wonderful palette of high-pressure techniques with which to investigate the physical properties of molecules. In combination this proffers an extremely powerful toolkit.

## Methods

**Synthesis of 1 and 2.** All manipulations were performed under aerobic conditions, using chemicals as received. Type 3 Å molecular sieves were used to dry the MeCN before use. Compound **1** was prepared following the literature procedure[32]. The synthesis of **2** was performed following an alternative method to that described in the literature and starting from $(NBu_4)_2[ReBr_6]$, which was previously prepared by a metathesis reaction of $K_2ReBr_6$ in a 0.5 M HBr solution[33]. A mixture of $(NBu_4)_2[ReBr_6]$ (0.15 g, 0.13 mmol) and 2,2′-bipyrimidine (0.02 g, 0.13 mmol) in 5 ml of glacial acetic acid was heated at 90 °C with continuous stirring for 2 h. It was then filtered and the obtained red–orange solution was left to evaporate

in a fumehood at room temperature. Dark red crystals of **2** suitable for X-ray diffraction studies were formed in 2 days. Yield: 40%. Analysis {calcd., found for $C_8H_6N_4Br_4Re$ (**2**)}: C (14.5, 14.3), H (0.9, 1.1), N (8.4, 8.5) %. IR peaks ($KBr/cm^{-1}$): 1,576(vs), 1,544(m), 1,404(vs), 814(m), 743(s), 665(m) for **2**.

**Structure determination and refinement.** Single-crystal X-ray diffraction data and high-pressure data were collected on a Bruker APEX II diffractometer with graphite-monochromated Mo–Kα radiation with a wavelength of 0.71073 Å for both compounds **1** and **2**. The high-pressure series collected on compound **1** was collected at station I19 at Diamond Light Source, using radiation of wavelength 0.48590 Å on a four-circle Crystal Logic diffractometer equipped with a Rigaku Saturn CCD detector. A Merrill–Bassett diamond anvil cell (half-opening angle 40°), equipped with Boehler–Almax diamonds with 600 μm culets and a tungsten gasket was used[45]. Pressure was measured before and after each data collection via ruby fluorescence, with compound **1** loaded in a 40:60 petroleum ether mixture and compound **2** in Daphne 7373 oil[46]. Cell indexing and data processing were carried out using the Bruker APEX II suite, with the Rigaku frames obtained in I19 being converted to Bruker compatible frames using the programme ECLIPSE[47]. Integration was carried out using SAINT, with dynamic masks generated by ECLIPSE. Absorption corrections were carried out with SHADE, to account for cell shading and with the routine non-empirical method of SADABS[48]. Structure solution and space group determination were carried out using SHELXT[49] for the new phase of compound **2**. The ambient in-cell data for compound **2** were collected after the pressure series. On decompression the crystal split and was integrated and refined as a twinned crystal. All other structures were modelled from the starting coordinates. Refinements were carried out for compound **1** using CRYSTALS (against $F$ using reflections with $I > 2\sigma$) and SHELXL through the Olex2 software suite for compound **2** (against $F^2$ using reflections with $I > 2\sigma$)[50]. All metal-ligand distances and angles, and all torsion angles were refined freely, with RIGU[51] thermal and vibrational similarity restraints applied throughout. The compressive indicatrices were calculated using Pascal[52].

**Magnetic studies.** Dc magnetic susceptibility and magnetization measurements at ambient pressure were carried out on a microcrystalline sample of **1** in a Magnetic Property Measurement System (MPMS, Quantum Design, USA) magnetometer equipped with a 7 T magnet operating in the 300–1.9 K temperature range. High-pressure dc magnetic susceptibility measurements were carried out in an MPMS SQUID equipped with a 5 T magnet on microcrystalline samples (~10 μg) of **1** and **2** (**1** @ ambient—4.1 GPa; **2** @ ambient—4.2 GPa) in the 50.0–2.0 K temperature range and an external magnetic field of 100 Oe in a turnbuckle diamond anvil cell with 800 μm culets, using ruby powder as the pressure calibrant, and Daphne 7,373 oil as the pressure transmitting medium. The cell and gasket were made from ultrapure CuBe 165. See ref. 34 for full details of the cell.

**Computational methods.** Full details of the computational methodology are given in the Supplementary Methods and Supplementary References.

**Data availability.** All data supporting the findings of this study are available from the authors. The X-ray crystallographic data (CIF files) for the structures reported in this work have been deposited at the Cambridge Crystallographic Data Centre (CCDC), under deposition numbers CCDC 966619–966625 (**1**), CCDC 1453399–1453406 (**2**) and CCDC 1491360–1491361 (ambient, $T = 4$ K for **1** and **2**). These data can be obtained free of charge from The Cambridge Crystallographic Data Centre via www.ccdc.cam.ac.uk/data_request/cif. All other data may be obtained from Edinburgh Datashare: http://datashare.is.ed.ac.uk/. Alternatively, data may be obtained from the corresponding authors upon request.

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

## Acknowledgements

Financial support of the EPSRC (EP/K033646/1, EP/K033662/1 & EP/N01331X/1) and the Spanish Ministerio de Economía y Competitividad (projects CTQ2013-44844-P and MDM-2015-0538 (Excellence Unit 'María de Maeztu')) and the Generalitat Valenciana (projects PROMETEO/2014/070 and ISIC/2012/002) is gratefully acknowledged. We acknowledge Diamond Light Source for time on Beamline I19 under Proposal MT11879.

## Author contributions

E.K.B., K.V.K., S.P., S.M., M.M. and M.R.P. conceived the idea for the project and obtained funding. J.M.-L and J.F. performed the synthesis. C.H.W. and A.P. carried out the high-pressure X-ray data collection and structure determination with supervision from S.M., S.P. and M.R.P. C.H.W., G.A.C., M.M., M.M. and K.V.K. designed and performed the high-pressure SQUID experiments and associated data analysis. J.C. performed the theoretical studies. E.K.B., M.M., G.A.C., C.H.W., J.M.-L and J.C. wrote the manuscript, which all authors discussed and commented on.

## Additional information

**Competing financial interests:** The authors declare no competing financial interests.

**Publisher's note**: 

