## [Peer Review File · Nature Communications]

REVIEWERS' COMMENTS:

Reviewer #1 (Remarks to the Author):

The revised manuscript included comprehensive and detailed answers to most of my previous concerns, and therefore I recommend publication in Nature Communications in the present form.

Reviewer #2 (Remarks to the Author):

In this manuscript, the authors combine high pressure structural and magnetic studies to probe the effects of structural distortion on the magnetic ordering of Re(IV) complexes. The changes in ordering temperature are correlated to the changes in inter-molecular distances and (J) and intramolecular anisotropies (D and E). Overall the experiments and analyses are interesting, the arguments are conceived well and presented clearly and logically, and the writing is superb. All in all it was a pleasure to read the manuscript, and in my view it is a high quality communication worthy of publication in a high-profile journal. In my original review, I was concerned with some of the literature citations, novelty, and some experimental details. In the revised paper, the data quality is improved and the low temperature structure is a good addition. The authors have addressed my concerns thoughtfully. I think the revised paper is ready for publication in Nature Communications.

Reviewer #4 (Remarks to the Author):

The manuscript submitted by Woodall and co-workers have been revised in an extensive and very professional manner, improving a lot the quality of the reported data (like for example the magnetic data shown in the figures 2 and 3), the writing of the manuscript in particular the introduction and also the supporting information. The referee is actually impressed about the efforts done by the authors to get the best possible paper. So the referee has not much to say on the scientific point of view, everything mentioned in his previous report to Nature Chemistry has been taking into account or considered by the authors. This is also true for the different points mentioned by the other referees. The reported science is of high quality and this work should be published as it is. Nevertheless, this manuscript is submitted now to Nature Communications and the referee should estimate if this work is suitable for an high ranked journal of the Nature series. In the present case, this is a very difficult point to judge as it depends on the referee feeling of what should be published in this journal. On a very personal point of view, the referee does not see a major breakthrough in this report (as written in the previous report: "no new compound, no new technique, no results with a broad/general impact...") that justifies its publication in Nature Communications. Being "the first combined high pressure structure-magnetism study of an ordered material, the first example to involve purely molecular species, and the first example involving 5d metal ions" is not something that is general enough or useful for a broad community of scientists. Of course this combination of techniques could be useful for studying molecular systems but as mentioned by the authors in their letter, "such pressure measurements are non-trivial and very time consuming - the dc magnetic data presented here, for example, required many months of SQUID time.". Therefore this is unlikely an scientific strategy that is not going to be used by many research groups in the future.

REVIEWERS' COMMENTS:

Reviewer #1 (Remarks to the Author):

The revised manuscript included comprehensive and detailed answers to most of my previous concerns, and therefore I recommend publication in Nature Communications in the present form.

Reviewer #2 (Remarks to the Author):

In this manuscript, the authors combine high pressure structural and magnetic studies to probe the effects of structural distortion on the magnetic ordering of Re(IV) complexes. The changes in ordering temperature are correlated to the changes in inter-molecular distances and (J) and intramolecular anisotropies (D and E). Overall the experiments and analyses are interesting, the arguments are conceived well and presented clearly and logically, and the writing is superb. All in all it was a pleasure to read the manuscript, and in my view it is a high quality communication worthy of publication in a high-profile journal. In my original review, I was concerned with some of the literature citations, novelty, and some experimental details. In the revised paper, the data quality is improved and the low temperature structure is a good addition. The authors have addressed my concerns thoughtfully. I think the revised paper is ready for publication in Nature Communications.

Reviewer #4 (Remarks to the Author):

The manuscript submitted by Woodall and co-workers have been revised in an extensive and very professional manner, improving a lot the quality of the reported data (like for example the magnetic data shown in the figures 2 and 3), the writing of the manuscript in particular the introduction and also the supporting information. The referee is actually impressed about the efforts done by the authors to get the best possible paper. So the referee has not much to say on the scientific point of view, everything mentioned in his previous report to Nature Chemistry has been taking into account or considered by the authors. This is also true for the different points mentioned by the other referees. The reported science is of high quality and this work should be published as it is. Nevertheless, this manuscript is submitted now to Nature Communications and the referee should estimate if this work is suitable for an high ranked journal of the Nature series. In the present case, this is a very difficult point to judge as it depends on the referee feeling of what should be published in this journal. On a very personal point of view, the referee does not see a major breakthrough in this report (as written in the previous report: "no new compound, no new technique, no results with a broad/general impact...") that justifies its publication in Nature Communications. Being "the first combined high pressure structure-magnetism study of an ordered material, the first example to involve purely molecular species, and the first example involving 5d metal ions" is not something that is general enough or useful for a broad community of scientists. Of course this combination of techniques could be useful for studying molecular systems but as mentioned by the authors in their letter, "such pressure measurements are non-trivial and very time consuming - the dc magnetic data presented here, for example, required many months of SQUID time.". Therefore this is unlikely an scientific strategy that is not going to be used by many research groups in the future.